# Correlation of Macroscopic Fracture Behavior with Microscopic Fracture Mechanism for AHSS Sheet

**DOI:** 10.3390/ma12060900

**Published:** 2019-03-18

**Authors:** Lingyun Qian, Xiaocan Wang, Chaoyang Sun, Anyi Dai

**Affiliations:** 1School of Mechanical Engineering, University of Science and Technology Beijing, Beijing 100083, China; w126xc@163.com (X.W.); qianlingyunabc@163.com (A.D.); 2Beijing Key Laboratory of Lightweight Metal Forming, Beijing 100083, China

**Keywords:** TRIP 780, fracture mechanism, stress state, stress triaxiality, numerical simulation

## Abstract

This research aims to correlate the macroscopic fracture phenomenon with its microscopic fracture mechanism for an advanced high-strength steel (AHSS) TRIP 780 sheet by applying a combined experimental-numerical approach. Six specimens with different shapes were tensioned to fracture and the main deformation areas of specimens were subjected to stress states ranging from lower to higher stress triaxiality. The final fracture surface feature for each specimen was obtained to characterize the macroscopic fracture modes at different stress states. The scanning electron microscope (SEM) fractographies of fracture surfaces were detected to reveal the microscopic fracture mechanisms. The stress triaxiality evolution was applied to correlate of fracture mode and fracture mechanism by comparing the macroscopic fracture features as well as micro-defect changes. An increase of stress triaxiality leads to voids extension and then results in a voids-dominant fracture. The micro-shear-slip tends to appear in the stress triaxiality level lower than that of pure shear stress state. The fracture behavior of a practice deformation process was the result of interplay between shear-slip fracture and void-dominant fracture. The unified relationship between average void sizes and stress triaxiality was obtained. The void growth was predicted by the Rice–Tracey model with higher precision.

## 1. Introduction

The automotive industry has witnessed a dramatic increase in the application of lightweight materials due to the mounting pressure of cost efficiency and fuel economy in recent years. These materials feature a superior strength to weight ratio and thus promise a substantial weight reduction. Among the promising lightweight materials of car bodies, TRIP steels as advanced high-strength steels (AHSS) have gained more attention due to their more mature techniques and relatively lower costs [1]. However, the flow strength increase of metals would create big challenges for material formability as well as fracture issues for manufacturing processes. The ductile fracture of metallic material happens when the underlying damage due to micro-defect evolution of severe plastic deformations accumulates to a certain extent. Therefore, to capture the microscopic mechanism of damage evolution and explain the correlation with macroscopic fracture phenomena is of great importance for optimizing the deformation quality of sheet metals.

In a microscopic view, the ductile fracture of metals is caused by the integral process of the nucleation, growth and coalescence of micro-defects including micro-voids and micro-shear-slip; localized necking and shear fracture are two general macroscopic fracture forms. The essence of fracture behavior is the evolutions of micro-defects which are governed by the stress states acting in material points. It is of great significance to establish a suitable fracture criterion to improve the prediction precision of fracture behavior, especially criteria which reflect fracture mechanisms.

Since Rice and Tracey [2] stated the effect of stress triaxiality on voids development and also established a function to present the relationship, stress triaxiality has become an important parameter to illustrate fracture behavior. Several criteria with different forms have been proposed on basis of this [3,4,5,6]. Bao and wierzbicki [7] divided the function curve of stress triaxiality and plastic strain into three branches and pointed out that the change trend at low stress triaxiality between 0 and 1/3 was ambiguous by checking the experimental and simulated results of several different specimens. As research continued, they also evaluated seven fracture criteria and only Tresca fracture model worked well to predict shear fracture [8]. This similar issue was also raised by Li et al. [9] after they checked several fracture criteria through some basic experiments and practical examples with diverse stress states. They illustrated that predictive ability of fracture criteria highly depend on whether they can accurately reflect the microscopic fracture mechanism. Recently, more and more attentions have been paid to characterize the shear fracture and lode parameter as the preferential characterizing variables have been incorporated into many fracture criteria [4,5,10,11,12] MMC fracture model has been applied by Li et al. [1], Qian et al. [13,14] and Luo and Wierzbicki [15] to successfully predict fracture under the stress states of in-plane shear, uniaxial tension and plane-strain tension during sheet metal forming processes. Lou et al. [16,17] proposed a new shear-controlled fracture criterion considering the ductile fracture mechanism of void evolution. One novel criterion KHPS has been introduced by Kubik et al. [18] to predict the failure at negative stress triaxiality using newly designed specimens. In recent years, research hotspots have focused on ways to improve the predictive precision in negative and lower stress triaxiality as well as to determine the cut-off value of stress triaxiality [19,20,21]. These current investigations mostly study the fracture behaviors by adopting fracture criteria proposed on basis of different failure modes on the macro-scales. Experiments and numerical modeling on the micro-scales are in great need in order to further reveal underlying fracture essence of metallic material.

Zhu et al. [22] discussed the fracture mechanism at different stress states by conducting in situ tensile tests in scanning electron microscopy (SEM) of 6063 aluminum alloy and numerical simulations. Normal stress together with shear stress were deemed to dominate final fracture modes. Lou et al. [23] observed the fracture surfaces of several experiments ranging from plane strain compression to biaxial tension to analyze the fracture mechanism and the maximum shear stress was brought forward to govern the ductile fracture and should be coupled into fracture criteria. The combination of fracture micro-mechanism (voids enlargement and coalescence, formation of micro-shear-cracks) and macroscopic variables (stress intensity, stress triaxiality, Lode parameter) were used to construct novel fracture models [20,24,25]. Besides the common method to check fracture sections by SEM, a thin metal sheet containing laser drilled holes was tensioned in SEM and the micro-defects evolution as well as microscopic fracture mechanism were determined on basis of hole configuration changes [26]. Synchrotron radiation-computed tomography was utilized to reveal voids evolution during crack propagation [27]. The microscopic evolution of the fracture mechanism is fairly difficult or even impossible to determine by experimental methods. The macroscopic stress states variables were applied as bridges to explain the microscopic fracture mechanism. In this framework, various stress-state-dependence fracture criteria and numerical simulations on basis of unit cell model have been performed [28,29,30].

These research works studied the evolution process of single or multiple micro-defects from initial state to accumulation leading to macro-cracks under various loading conditions. This makes it possible to reveal the fracture mechanism that cannot be reflected by experiments. For example, the micromechanical model containing a three-dimensional unit cell was developed by Barsoum and Faleskog [28] to analyze the effects of the lode parameter on void growth and coalescence. Brünig et al. [31] conducted numerical simulations on the micro-scale using three-dimensional unit cell models, and macro-scale experiments have also been performed to better understand the complex stress-state-dependent damage behavior.

Although this research concentrated on gaining more insight in the complex fracture behavior as well as underlying microscopic mechanism by experiments, SEM observation, micro-scale unit cell model and macro-scale numerical simulation, the research findings are diverse and make it difficult to establish a unified fracture model with a wide range of engineering applications. Most of the current phenomenological fracture models do not have physical significance and are also not related to fracture microstructures, therefore the macroscopic fracture behaviors are rarely analyzed on the basis of a microscopic fracture mechanism controlled by deformation modes. The correlation between fracture modes and macroscopic fracture mode has not been fully investigated. Li et al. [9] presented a rough discussion on the correlation between micro fracture modes and deformation modes and did not explain the macroscopic fracture modes in terms of specific stress states.

Therefore, this research focuses on the correlation of macroscopic fracture phenomena with microscopic underlying mechanisms of diverse stress states for a TRIP 780 sheet. Experiments and numerical simulations on the macro-scale are performed for five specimens to obtain shape features and stress triaxiality evolution of fracture surfaces. SEM of fracture surfaces of these tests is applied to analyze the macroscopically and microscopically with aims to relate the micro-defects characteristics and stress triaxiality evolution. Thus, the fracture mode and corresponding mechanisms at different stress states are elucidated.

## 2. Experimental and Simulation Conditions

### 2.1. Experimental Procedure

The TRIP 780 sheet with a thickness of 1.0 mm was selected as test material and it was annealed before tests to achieve the relatively higher ductility. In order to study the fracture mechanism at different stress states, five specimens were designed to have different notch shapes where different stress states could be induced at these localized deformation zones. The main stress states range from shear-dominated to tension-dominated loading conditions, as seen in Figure 1. The left three shear-dominated specimens was named as S-0, S-30 and S-45 according to the expected angles between fracture direction and tension direction during tests. The Specimen S-0 was designed symmetrically to decrease the effect of rotation on the shear stress state of an unsymmetrical specimen. The other two notched specimens ware called as NT-R9 and NT-R3 according to their notch radii. Slow-feeding wire cutting ensured a high accuracy of machining to avoid the initial edge imperfections which might result in accidental fracture onset.

All tension tests were conducted on a universal material testing machine under displacement control at a constant crosshead velocity of 1 mm/min. Each specimen was tested twice to ensure the repeatability of results. The digital correlation image (DIC) method was used to calculate the strain information during tests as well as to verify the numerical modelling. The localized fracture zones of all specimens were cut and corresponding fracture surfaces were observed by SEM after ultrasonic cleaning.

### 2.2. Finite Element (FE) Modeling

The parallel 3D finite element (FE) models of six tests were built on the platform of ABAQUS/Explicit. The specimen S-0, S-30 and S-45 adopted a symmetrical model along the thickness direction. Considering the symmetrical load and geometry of the tension-dominated specimen, one-eighth of the specimen was adopted in the FE models for specimen NT-R9 and NT-R3. All specimens were meshed by eight-node hexahedral elements C3D8R with reduced integration. The smallest mesh size was 0.05 mm. That is to say, twenty elements were meshed in the thickness. The material was defined as elastic-plastic with a Young’s modulus of 210 GPa and Poisson ratio of 0.33. The relationship between true stress and plastic strain was shown in the Figure 2 and it was determined by an FE-based inverse engineering method combined with experimental results [13].

## 3. Results and Discussions

### 3.1. Macroscopic Fracture Behavior

Figure 3 shows the fracture morphology of five deformed specimens with various shapes. As shown in the figure, the final fracture regions located in the pre-designed notching-areas and different certain degree of necking appeared in the thickness direction for all specimens. For three shear-dominated specimens of S-0, S-30 and S-45, the angles between fracture surfaces and experimental tension direction gradually increased from 0° to 45°, which corresponded with the expected results. The discrepancies between actual and expected values were caused by uncertain experimental errors. For two notched specimens NT-R9 and NT-R3, the fracture regions emerged with significant necking in both width and thickness directions and the fracture surfaces were almost perpendicular to the tensile direction.

Simulated and measured load-displacement curves for five specimens were shown in Figure 4. It is clear that the results obtained from the FE simulations agreed well with the experimental measurements. Among three shear-dominated specimens, S-0 had the maximum peak load of 4600N and largest fracture displacement of 3.2 mm. With the increase of fracture angle, the fracture displacement has shown a decreasing trend and the fracture stroke of specimen S-45 was about 2.2 mm. However, the peak loads almost stayed at the same level of 3000 N for specimen S-30 and S-45. For notched specimens of NT-R3 and NT-R9, the force-displacement curves varied with similar trends. The NT-R3 had a larger peak load of 5400 N and a smaller fracture stroke of 2.1 mm. It seems that specimens with smaller notch radius were prone to emerge earlier in fractures under tension loading.

Besides the global force-displacement curve, the local logarithmic axial strain of certain point was served as another main evaluation index to verify the robustness of the present FE models. Specimen No. 2 was selected as an example to demonstrate how to verify the FE models. Figure 5 shows the evolution of strain along tensile direction with a displacement for the central point. The black dotted line denoted experimental results calculated by using the DIC method and red dotted line denoted simulation results. The measured and calculated strains were basically identical, which means that the present FE simulation with determined material properties had a high precision. Therefore, conclusions could be drawn that the established material properties and finite element models are reliable and can be applied for further analysis.

### 3.2. Stress Triaxiality Evolution

The macroscopic fracture mode and microscopic fracture mechanism both depend the stress states during deformation for a metallic material. There are various ways to describe and quantify stress states. Among these evaluating parameters, stress triaxiality is more comprehensive and widely used by many researchers [4,7,16,32]. Stress triaxiality is defined as the ratio of mean stress to von Mises equivalent stress and is considered to affect the growth of micro voids and therefore the final fracture behaviors. Hence, it is also widely used to construct fracture criterion. The evolutions of stress triaxiality of fracture surface along width and thickness directions were abstracted from numerical results for different specimens to analyze the evolution principles of stress states. Figure 6 shows the stress triaxiality along the width of the middle layer of the fracture surface and the start-stop points of the path was marked by the red arrow.

The levels of stress triaxiality within the central region for two notched specimens NT-R3 and NT-R9 were higher with maximum values more than 0.55, as shown in Figure 6a. The stress triaxiality gradually decreased from the center to edges and the distribution curves were close to “conical cups”. The total levels of stress triaxiality of specimens NT-R3 and NT-R9 were larger than the analytical value of 1/3 of uniaxial tension stress state, which indicated that these two notched specimens underwent tension-dominated deformation with a relatively higher stress triaxiality. Besides, the curve of stress triaxiality of specimen NT-R3 was higher than that of specimen NT-R9. That is to say, the stress triaxiality of the notched specimen increased with the decrease of notched radius. As discussed in Figure 4, the specimen NT-R3 was prone to occur fracture earlier than specimen NT-R9. It can be concluded that the higher level stress triaxiality of notched specimens could accelerate fracture occurrence.

Figure 6b shows the stress triaxiality along the fracture path of specimen S-0. The curve between the central region divided by two dashes changed gently and the values fluctuated around 0.16. The value 0.16 is larger than the analytical value zero of the pure shear stress state, which indicated that the pure shear stress state was difficult to achieve in practice. Therefore, the central deformation region was mainly governed by the shear-dominated stress state. It can be seen that the stress triaxiality became larger from center to edge and reached its peak value of 0.45. This shift of stress triaxiality level shows that the primary stress states during deformation converted from shear-domination to tension-domination. Figure 6c shows the outcomes of the specimens S-30 and S-45. The stress triaxiality along fracture path for specimen S-30 was around 0.20. Therefore, shear-dominated stress state affected the fracture behavior for specimen S-30. Compared with specimen S-30, the stress triaxiality increased to about 0.29 with the level close to 1/3 corresponding to uniaxial tension stress state. The results indicated the effect of tension-dominated stress state are gradually highlighted with the increasing angles between fracture path and tension loading direction for shear-dominated specimens.

### 3.3. Microscopic Fracture Mechanism

Figure 7 displays the fracture morphology of specimen S-0. The surface of the fracture was smooth from the overall topography and the fracture at the edge of the notch was rougher than central region. It can be seen from Figure 6b that its stress triaxiality in the central region was about 0.16, which indicated that the shearing effect came to predominate and the necking phenomenon was not obvious. The stress triaxiality of edges was increased and the maximum value reaches 0.45. The greater stretching effect resulted in increasing necking appearance. Therefore, it can be seen that the section of specimen S-0 was thick in the middle and thin on both sides from the overall shape. The surface of the fracture morphology at the center region along the thickness direction contained a large number of shear planes and only a small number of shallow and small dimples with sizes of less than 1 μm. The surface of fracture morphology at edge region in thickness direction was still a large number of shear planes. At the same time, the stretching direction of dimples at the center and edge in thickness direction was nearly similar with the stretching direction, as indicated by blue arrows in Figure 7. The differences between central and edge part was that the amount of voids at edges increased due to the gradually increasing stress triaxiality as shown in Figure 6b. Therefore, the distributions of voids in the shear bands became a little disorderly.

Figure 8 shows stress triaxiality along the center line of the minimum cross-section (marked by red arrow) in thickness direction for specimen S-0. The stress triaxiality had a lower level with the maximum value of 0.16. The reduction of stress triaxiality from center to edges indicated that the shear effect was even stronger. That is why the sizes of voids near edge were smaller than that of the center region, as depicted in Figure 7. In consequence, the shear-dominant fracture characterized mainly by shear slipping bands occurred for the center region of specimen S-0.

Figure 9 plots the fracture morphology of two shear-dominated specimens S-30 and S-45. As a whole, their surfaces were both rougher than that of specimen S-0. Compared with specimen S-0, their fracture has an appearance with numerous voids in the central interior areas and several shear bands with minor voids near the boundary. This fact can be seen from the stress triaxiality distributions on minimum cross-sections along the thickness direction of two specimens as shown in Figure 10. The distribution curves were close to symmetrical parabolas with the maximum values in the central positions. The stress triaxiality of specimen S-30 was larger than S-0 but smaller than S-45. Thus, the overall size of voids in the central region of specimen S-45 was 4 μm and larger than the value 3 μm of specimen S-30. The similar phenomenon was also detected for the marginal areas. The peak value of stress triaxiality for specimen S-45 was about only 0.26. The stress triaxiality levels along the thickness and width (seen in Figure 6c) of these two specimens lay in the lower stress triaxiality between 0 and 1/3 where the evolution of shear-ship defects is the dominant microscopic fracture mechanism.

Figure 11 shows the SEM fracture fractographies of the two notched specimens NT-R3 and NT-R9. Obvious necking was observed in both width and thickness direction of macroscopic fracture sections. Different quantities of voids were observed in the center regions as well as the edge regions. Bits of slipping bands could be also checked among the voids, especially in edge regions, as shown in Figure 11. On the whole, in specimens NT-R3 and NT-R9 void-dominant fracture occurred with a higher level of stress triaxiality shown in Figure 6a. Compared with the SEM fracture fractographies of three shear specimens shown in Figure 7 and Figure 9, the distribution pattern of voids were disordered for the two notched specimens. These microscopic mechanism were caused by different stress states, as depicted by evolutions of stress triaxiality in Section 3.2. In the specimen NT-R3 (Figure 11a), the size of enlarged voids was in the order of 3–5 μm and in a few cases voids with sizes of 5 μm were detected. The void sizes were smaller and in the order of 2–3 μm for specimen NT-R9. The smaller notch radii and corresponding higher stress triaxiality accelerated the growth of voids as well as coalescence, and thus materials tended to become weaker for fractures. With the reduction of notch radius, the influence of stress triaxiality on voids development together with tension-dominated stress states became more prominent. Figure 12 demonstrates the stress triaxiality along the thickness of specimen NT-R3 and NT-R9. All values of stress triaxiality were greater than 1/3 and decreased from the center of the specimens toward the boundaries. The tension stress states are dominated during the deformations of two specimens. However, the tension-dominated effects gradually weaken from center to edge. That is why a few slipping bands were detected around boundaries. Above all, stress triaxiality of the notched samples was greater than 1/3 and the fracture mode was a void-dominated fracture.

Specimen NT-R3 was selected to state the micro-topography changes along the thickness for notched specimens. As shown in Figure 12, three regions A, B and C were marked from the center towards the boundary. It can be seen that, the enlarged voids were distributed in large area of region A. The size of voids was mainly around 4 μm, whereas the region B was full of voids with a smaller size of 2 μm. A small amount of shear slips can be seen around the voids. In region C, which is close to the edge, the proportion of shear slip (marked by red arrows) increased and only rare voids existed. These micro-topography change and corresponding feature evolutions were caused by different stress states. The fracture mechanism changed from void-dominant fracture to shear-slip dominant fracture. It is difficult for the ideal fracture mechanisms, whether voids fracture or shear slip fracture, to exist individually in practical deformation. The real fracture process is a result of the interplay between shear-slip fracture and void-dominant fracture.

### 3.4. Analytical Relationship between Void Size and Stress Triaxiality

The essence of ductile fracture is the evolution behavior of micro-voids. It is generally recognized that the stress triaxiality is one dominant parameter to govern the void growth as well as coalescence. There exist many models to evaluate the relationship between stress triaxiality and the radius of voids on the fracture surface. The Gurson model and its extended expressions can better reflect the microscopic deformation mechanism and have been widely incorporated into several software programs to describe the micro-void’s evolution. However, this involves many parameters which are difficult to validate using traditional tests and even defined by resumption. There is little research to characterize the relationship between void size and stress triaxiality. The energy-type models, such as the CDM model and Rousselier model, use macroscopic parameters to describe the damage evolution process of metal materials. These models normally present implicit analysis of micro-void size changes and are not easy to validate directly by using experimental data. Compared with the above two types of models, the Rice–Tracey model has relatively simple expression and is widely used by many researchers to analyze void size evolution. Therefore, it was utilized in this research. In the Rice–Tracey model, Critical voids growth ratio (*R*/*R*_0_)*_c_* can be written as [33]:(1)$$\ln{\left(\frac{R}{R_{0}}\right)}_{c}=\int_{\varepsilon_{0}=0}^{\varepsilon_{c}}0.283\cdot \exp\left(\frac{3\sigma_{m}}{2\sigma_{eq}}\right)d\varepsilon_{eq}$$
where *R* stands for the actual mean void radius, *R*_0_ is its initial value and was assumed to 1 μm here. *σ_m_*/*σ_eq_* represents stress triaxiality *η*, and *ε_eq_* is the equivalent plastic strain. *ε_c_* is the critical value of equivalent plastic strain at initial fracture moment. In this expression, stress triaxiality *η* changes with the equivalent plastic strain and, therefore, it is necessary to determine the evolution of stress triaxiality before calculate *R*. The values of stress triaxiality and equivalent plastic strain at critical fracture points on fracture section for six specimens were abstracted, as shown in Figure 13. It appears that the stress triaxiality varied within a narrow range for most of the specimens. Although theoretically the values should remain unchanged, in practice a constant stress state is not easy to maintain, even in the numerical simulations. Since the fracture initiation is an evolving process, adopting an average value can include the load history effect. The integral mean values were calculated by using the formula in Reference [13] and are marked in Figure 13.

The average void sizes around central points on fracture sections for six specimens were roughly measured and their radii were listed as measured void sizes (marked by *R*_exp_) in Table 1. It can be concluded that increasing stress triaxiality promoted voids growth for the tension-dominated range with stress triaxiality larger than 0.33. From specimen S-0 to specimen DB, the interaction of shear and tension effect initially increase void sizes but the trend of growth weakened again, as indicated by the void sizes of specimen S-45 and DB. This variation tendency of void size with stress triaxiality coincided well with that of fracture strain, as indicated by Figure 14. This phenomenon was also manifested in reference [16,17,34]

The relationship between stress triaxiality and equivalent plastic strain for each specimen in Figure 13 was taken into Equation (1) and then the corresponding void radius *R*_sim_ was calculated. The calculated results of six specimens are summarized in Table 1. The relative error between measured and calculated radius was defined as
(2)$$\Delta =\frac{R_{\exp}-R_{\mathrm{sim}}}{R_{\exp}}\times 100\%$$

It can be seen from Table 1, all errors of six specimens were lower than 15% and the biggest error of 13% appeared at specimen S-0 with stress state closest to pure shear. Despite the fact that the Rice–Tracey model was proposed on the basis of the ductile enlargement of voids in the triaxial stress field, especially for stress states with increasing hydrostatic tension, it can predict the void radii well at different stress states ranging from lower to higher stress triaxiality in this research.

## 4. Conclusions

Several macro/micro-scaled plastic deformation cases covering diverse stress states had been carried out to correlate the macro-scaled fracture phenomena with micromechanics for an AHSS TRIP 780 sheet by applying a combined experimental-numerical approach. The main conclusions drawn were as follows:Obvious necking localization occurred in two notched specimens of NT-R3 and NT-R9. The corresponding void-dominated fracture has a macroscopic appearance with numerical voids in the central area of the fracture area and an oblique appearance at the limited edge area, whereas in the shear specimen S-0 very minor voids appeared on the smooth fracture surface and therefore shear fracture occurred. As designed intermediate shear-type specimens of S-30 and S-45, the corresponding fracture phenomena changed with a feature characterized by an increasing number of voids with growing size as well as decreasing shear-slip appearance.The stress triaxiality influenced deformation mode (tension-dominated deformation, shear deformation, shear-dominated deformation) and thus induced various fracture mechanisms (void-dominated fracture, shear-slip dominated fracture). An increase in stress triaxiality (especially higher than 1/3) enhanced voids growth effect and made voids-dominated fracture with significant necking occur easily. With the decrease of stress triaxiality from 1/3 to 0, the micro-fractography of shear-slip increased and the fracture mode gradually shifted from void-dominated fracture to shear fracture.Increasing stress triaxiality promoted voids growth for the tension-dominated range with stress triaxiality larger than 0.33. For the stress triaxiality range 0.11(specimen S-0) to 0.33 (specimen DB), the interaction of shear and tension effect initially increased void sizes from 1 μm to 1.9 μm but the trend of growth weakened again. This variation tendency of void size with stress triaxiality coincided well with that of fracture strain. The void radius of different stress states can be well predicted by the Rice–Tracey model with all relative errors lower than 13%.

## Figures and Tables

**Figure 1 materials-12-00900-f001:**
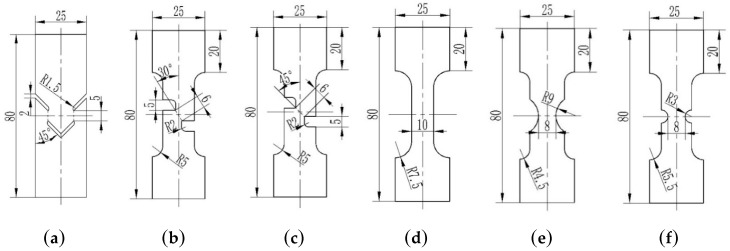
The geometry of five specimens (mm). (**a**) S-0; (**b**) S-30; (**c**) S-45; (**d**) DB; (**e**) NT-R9; (**f**) NT-R3.

**Figure 2 materials-12-00900-f002:**
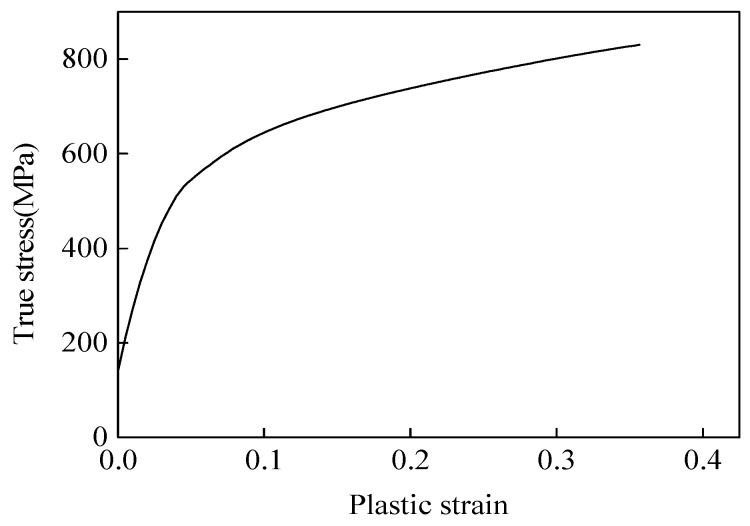
The true stress and plastic strain curve of TRIP 780.

**Figure 3 materials-12-00900-f003:**
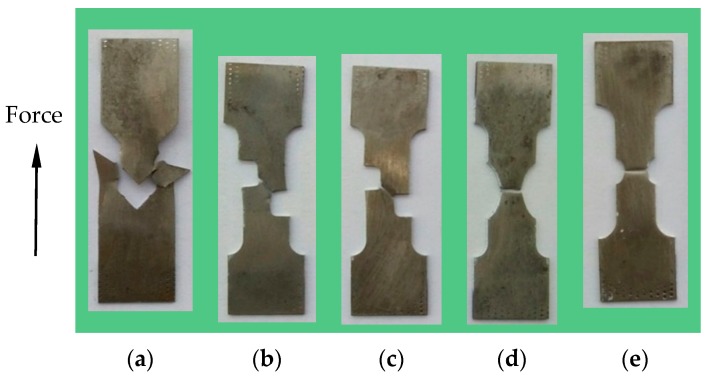
Macroscopic fracture morphology of five specimens. (**a**) S-0; (**b**) S-30; (**c**) S-45; (**d**) NT-R9; (**e**) NT-R3.

**Figure 4 materials-12-00900-f004:**
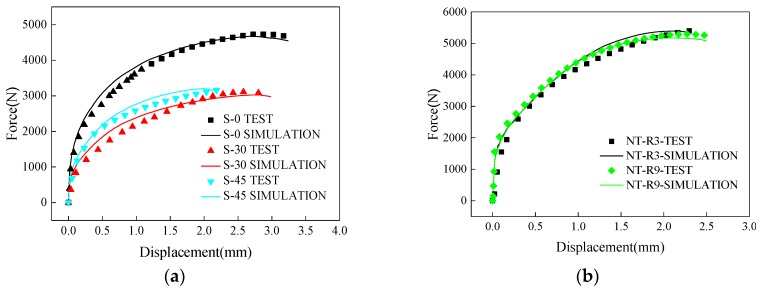
Comparison of experimental and simulated force-displacement curves. (**a**) Shear-dominated specimens; (**b**) notched specimens.

**Figure 5 materials-12-00900-f005:**
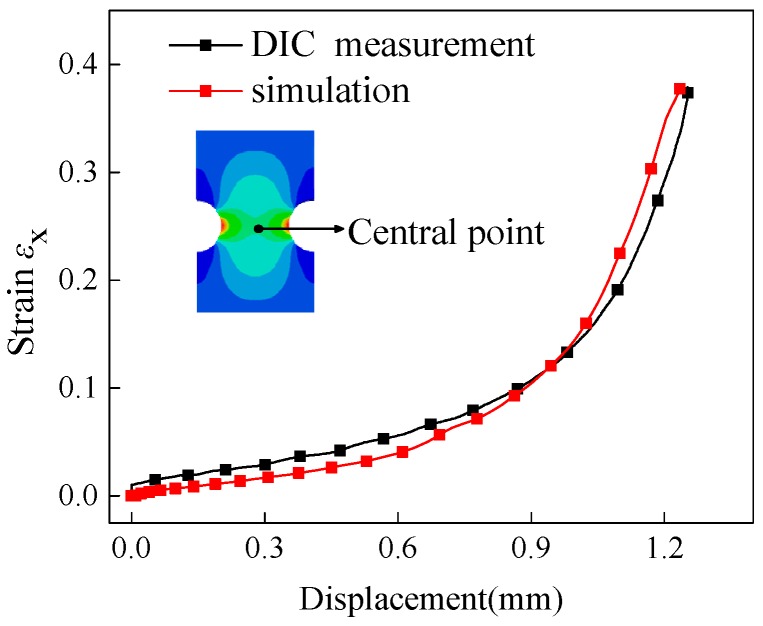
The strain evolutions measured by the digital correlation image (DIC) method and calculated by a finite element (FE) model for surface center point of specimen NT-R3.

**Figure 6 materials-12-00900-f006:**
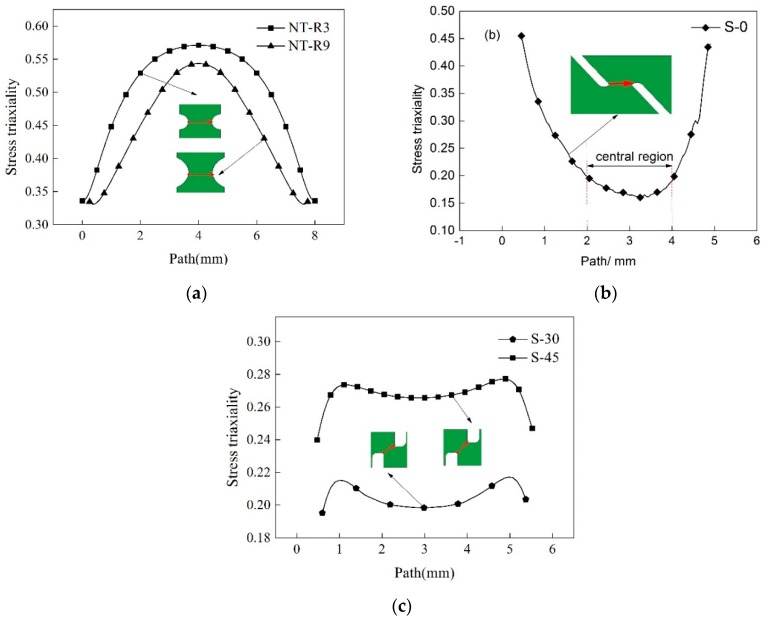
Stress triaxiality along width direction of five specimens. (**a**) NT-R3 and NT-R9; (**b**) S-0; (**c**) S-30 and S-45.

**Figure 7 materials-12-00900-f007:**
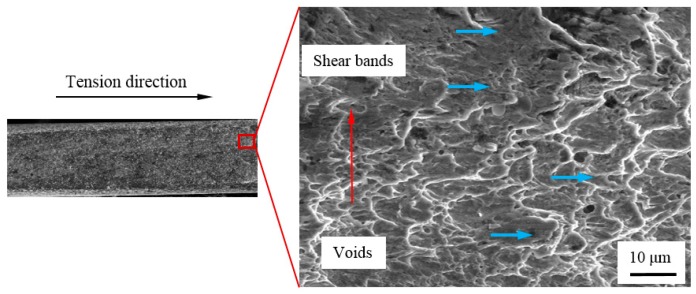
Fracture morphology of S-0.

**Figure 8 materials-12-00900-f008:**
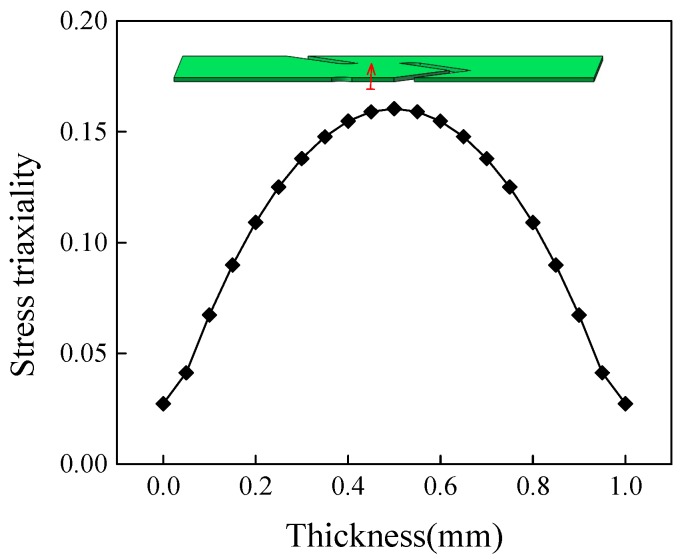
Stress triaxiality along center line of minimum cross-section in thickness direction for specimen S-0.

**Figure 9 materials-12-00900-f009:**
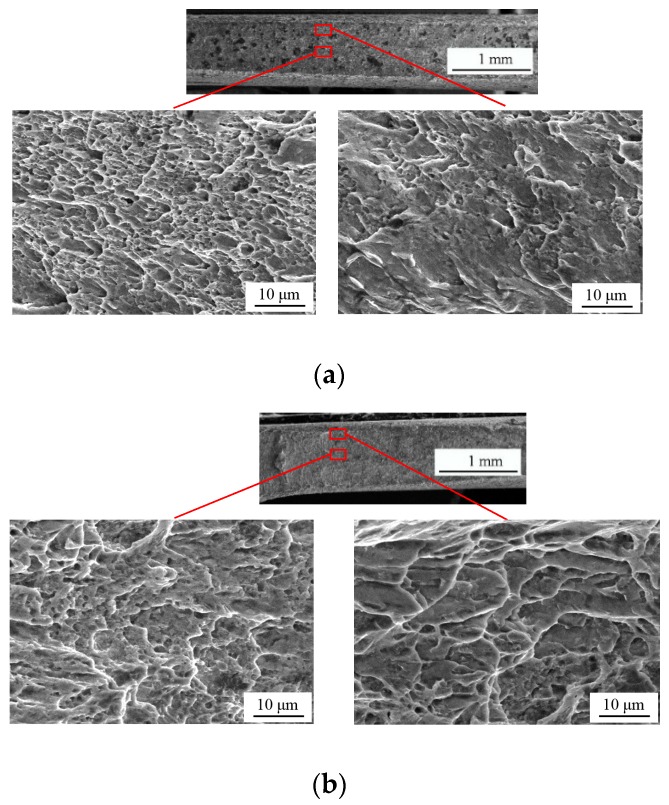
Fracture morphology of shear-dominated specimens: (**a**) S-30 and (**b**) S-45.

**Figure 10 materials-12-00900-f010:**
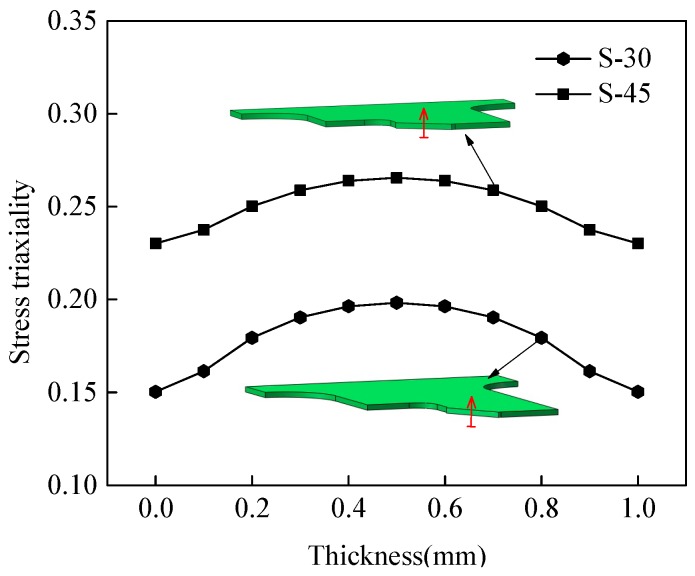
Stress triaxiality along center line of minimum cross-section in thickness direction for shear-dominated specimens.

**Figure 11 materials-12-00900-f011:**
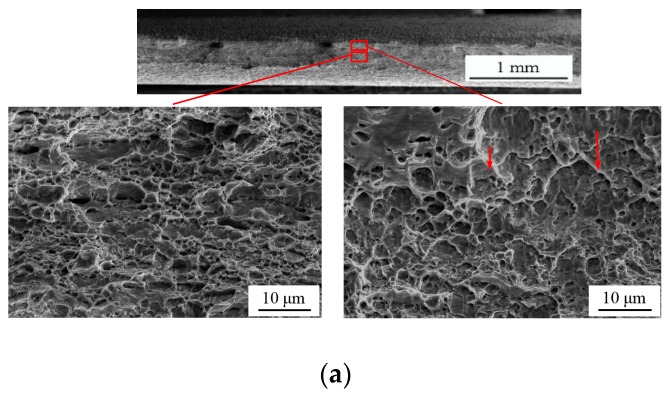
Fracture morphology of notched specimens: (**a**) NT-R3 and (**b**) NT-R9.

**Figure 12 materials-12-00900-f012:**
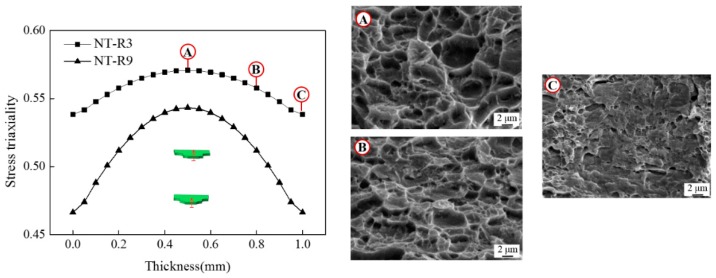
Stress triaxiality along thickness direction of notched specimens.

**Figure 13 materials-12-00900-f013:**
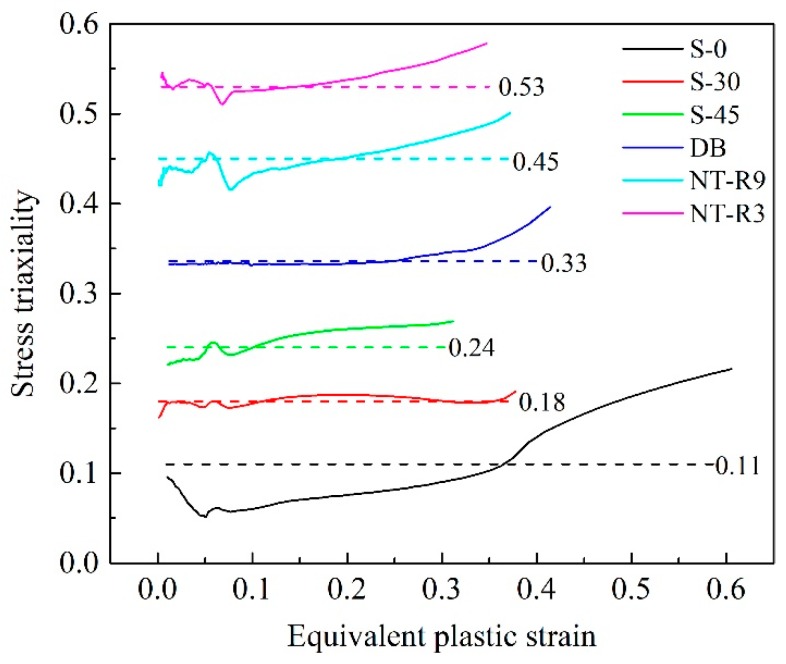
Evolution of stress triaxiality and integral mean value at fracture onset for six specimens.

**Figure 14 materials-12-00900-f014:**
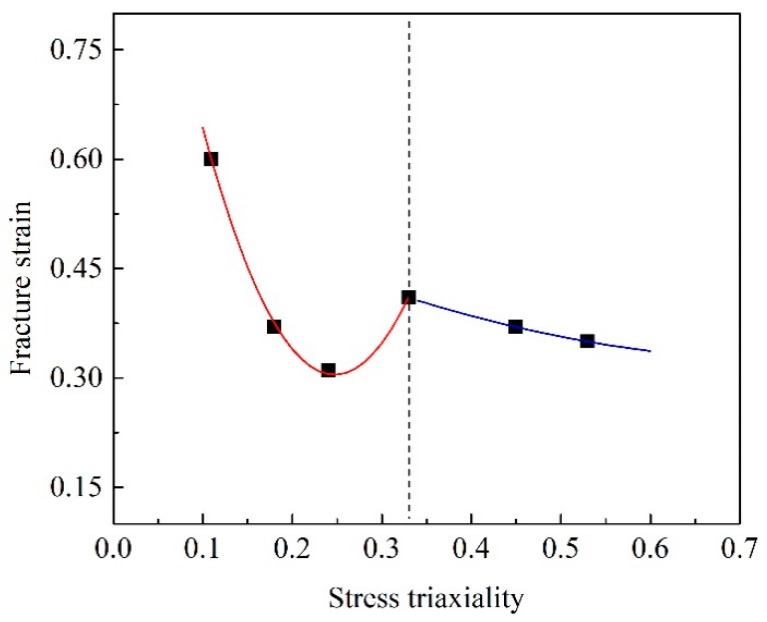
The relationship between stress triaxiality and fracture strain.

**Table 1 materials-12-00900-t001:** Stress triaxiality and measured/calculated void radius of the Rice–Tracey model.

Item	S-0	S-30	S-45	DB	NT-R9	NT-R3
Stress triaxiality	0.11	0.18	0.24	0.33	0.45	0.53
Measured void radius *R*_exp_ (µm)	1.00	1.71	1.90	1.55	1.71	2.13
Calculated void radius *R*_sim_ (µm)	0.87	1.58	1.79	1.50	1.78	1.99
Relative error ∆ (%)	13.00	7.60	5.62	3.23	−4.09	6.57

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
