# Peer review of "Correlation of Macroscopic Fracture Behavior with Microscopic Fracture Mechanism for AHSS Sheet"

_materials, 2019, doi:10.3390/ma12060900_

Reviewer 1 Report

This manuscript shows the new test results of 5 plate specimens having different stress triaxialities and has some discussions on fracture surface morphologies with the support of FEA simulations and SEM observations. The new test results might have a value in this research field but it might be difficult to find novel points in the results in its state. Therefore the recommendation is “mandatory revisions needed”. The followings are the major revised points raised by this reviewer.

In some places of the body of the text, the expression “Fig. * shown…” can be seen. It might be better as “Figure * shows…”.

There is little quantitative assessment results for fracture behaviors. Please address the quantitative discussion on:

・the unified relationship between average void sizes and stress triaxialities for 6 specimens including uniaxial- tension smooth specimen with the support of FEA and SEM.

・the range of stress triaxiality exhibiting the shear band dominant fracture surface

・the assessment of void size with the support of void nucleation model and void growth model, something like the Gurson model, Rousselier model and Rice&Tracy model or construct new void (and shear band growth) model agreeing with experimental bahavior 

Why did the authors use Abaqus explicit without using Abaqus standard? Can the accuracy of explicit FEA be shown?

Author Response

Thank you for kind contribution to improve our research. 

Detailed responses to the reviewer have been upload as a Word File, please check it.

Reviewer 2 Report

Overall, this is an interesting article with practically useful results for further research in this area. 

Results and Discussion section can be improved bringing-in more clarity.

Conclusions section can be improved highlighting the most important  findings in this work Vs. other peer's work.

Author Response

Thank you for kind contribution to improve our research. 

We have made major revision and added one new section including the following contents: 

1) Quantitative analysis of relationship between stress triaxiality and corresponding void sizes; 

2) Obtained the change trend of fracture strain at different stress states with varied stress triaxiality;

3) Adopted Rice-Tracey model to predict the void growth for all specimens.

The abstract and conclusion were improved by deleting non-key points and adding the new  breakthrough of revised contents.

Besides, we have made a careful check into the full text to avoid spelling mistakes and confusion. Some small spelling and typographical errors have been corrected in the revised manuscript. We also reorganized some sentences to enhance the readability of our paper. Not every revision has been marked, but some important changes have been highlighted in blue in the revised manuscript.

Round  2

Reviewer 1 Report

Thank you for the revision for my comments.

My recommendation is "publish as is".